



# Improved cloud detection over sea ice and snow during Arctic summer using MERIS data

Larysa Istomina[1,*], Henrik Marks[*], Marcus Huntemann[1,2], Georg Heygster[2] and Gunnar Spreen[2]

[1]Alfred-Wegener-Insitute, Helmholz Zentrum für Polar und Meeresforschung, Bremerhaven, 27570, Germany
[2]Institute of Environmental Physics, University of Bremen, Bremen, 28357, Germany
[*]Formerly at the Institute of Environmental Physics, University of Bremen, Bremen, 28357, Germany

*Correspondence to*: Larysa Istomina (larysa.istomina@awi.de)

**Abstract.** The historic MERIS sensor onboard Envisat (2002-2012) provides valuable remote sensing data for the retrievals of the summer sea ice in the Arctic. MERIS data together with the data of recently launched successor OLCI onboard
Sentinel 3A and 3B (2016 onwards) can be used to assess the long-term change of the Arctic summer sea ice. An important prerequisite to a high-quality remote sensing dataset is an accurate separation of cloudy and clear pixels to ensure lowest cloud contamination of the resulting product. The presence of 15 VIS and NIR spectral channels of MERIS allow high quality retrievals of sea ice albedo and melt pond fraction, but make cloud screening a challenge as snow, sea ice and clouds have similar optical features in the available spectral range of 412.5 - 900nm.

In this paper, we present a new cloud screening method MECOSI (MERIS Cloud screening Over Sea Ice) for the retrievals of spectral albedo and melt pond fraction (MPF) from MERIS. The method utilizes all 15 MERIS channels, including the oxygen A absorption band. For the latter, a *smile* effect correction has been developed to ensure high quality screening throughout the whole swath. Three years of reference cloud mask from AATSR (Istomina et al., 2010) have been used to train the Bayesian cloud screening for the available limited MERIS spectral range. Whiteness and brightness criteria as well
as normalized difference thresholds have been used as well.

The comparison of the developed cloud mask to the operational AATSR and MODIS cloud masks shows a considerable improvement in the detection of clouds over snow and sea ice, with about 10% false clear detections during May-July and less than 5% false clear detections in the rest of the melting season. This seasonal behaviour is expected as the sea ice surface is generally brighter and more challenging for cloud detection in the beginning of the melting season.

The effect of the improved cloud screening on the MPF/albedo datasets is demonstrated on both temporal and spatial scales. In the absence of cloud contamination, the time sequence of MPFs displays a greater range of values throughout the whole summer. The daily maps of the MPF now show spatially uniform values without cloud artefacts, which were clearly visible in the previous version of the dataset.

The resulting cloud mask for the MERIS operating time, as well as the improved MPF/albedo datasets are available as swath
data and daily means on the ftp server of the University of Bremen https://seaice.uni-bremen.de/data/meris/gridded_cldscr/.

## 1 Introduction

No other surface type of satellite imagery has the unique features of bright reflecting, white snow surface. The task of snow detection therefore would be an easy task in the absence of clouds. However, the snow spectral signature (e.g. Warren, 1982) is also a feature of water and especially of ice clouds (Kokhanovsky, 2006). Possible snow impurities, snow grain size

differences, and liquid water content create fine differences between many snow types (Warren, 1982), but in general the spectra of snow and cloud are similar in the VIS and NIR, with the difference occurring beyond 1µm (e.g. channels at 1.6, 3.7, 11 and 12 µm).

For MERIS data with a spectral range from 412.5nm to 900nm, cloud detection over snow and sea ice a challenging task. Besides cloud screening for the remote sensing retrievals using MERIS data, correct cloud detection from MERIS in the

Arctic region may be important for synergy with the other sensors onboard ENVISAT, e.g. as an accurate cloud fraction for the hyperspectral sensor of coarser spatial resolution SCIAMACHY.

Depending on the retrieved parameter and sensor, the effect of a compromised cloud screening may be moderate (albedo and grain size, SGSP, Wiebe et al., 2013) to drastic (aerosol retrieval, Istomina et al., 2011; melt pond fraction retrieval, Zege et al. 2015). The retrievals of MPF and albedo discussed in this work misinterpret the cloud contamination as melting sea ice

surface which cannot be distinguished from the true melting surface and overlays the true values in the daily and weekly means. The resulting MPF and albedo datasets are thus strongly affected by the residual cloud contamination. The objective of this work is to resolve this issue by means of a better cloud discrimination and to provide MPF, albedo and cloud mask datasets of a better quality.

### 1.1  Available cloud screening approaches

Some sensors are better suited for the task of cloud screening but are not suitable for the given retrieval due to other limitations. E.g. the MODIS cloud mask (Ackermann et al., 1998; Liu et al., 2004) is one of the most comprehensive classification algorithms, however, as the MODIS sensor experiences saturation in some of the visible bands (Madhavan et al., 2012), it is impossible to use these data for the given sea ice albedo and melt pond fraction retrieval (Zege et al., 2015). As the MERIS sensor onboard ENVISAT does not have these limitations, it has been chosen for the retrievals of MPF and

albedo. MERIS is located on the same platform as AATSR and SCIAMACHY, contains an Oxygen A band, and provides total Arctic coverage every three days with its swath width of 1150km. Synergy of AATSR and MERIS is used in this work to train and test the developed cloud screening routine.

Three basic cloud screening approaches applicable to a spectroradiometer data can be distinguished among the available algorithms:

- *Analysis of time-sequences* of data, under the assumption that the short-term changes of the scene can be only introduced by clouds (e.g. Key and Barry, 1989; Diner et al., 1999; Lyapustin et al., 2008; Lyapustin and Wang, 2009; Gafurov and Bárdossy, 2009). Such an approach assumes surfaces with a constant and pronounced structure (Lyapustin et al., 2008;

Lyapustin and Wang, 2009). Although the approach proved to be effective for various natural and artificial surfaces, it is not applicable within this work due to the fast-evolving nature of melt ponds and the sea ice.

- Applying a *reflectance or brightness temperature absolute threshold* or their combination, e.g. ratio of reflectances in the form of NDVI. In this case, only a few channels are used (e.g. Minnis et al., 2001; Bréon and Colzy, 1999; Lotz et al., 2009;

Allen et al., 1990; Spangenberg et al., 2001; Trepte et al., 2001). The optical properties of snow in the VIS show weak spectral dependency. In the NIR and IR, however, the snow spectrum shows the typical "snow signature", i.e. values decreasing due to water absorption in the NIR, which also causes the dependence on the snow grain size due to different pathlength and absorption in the grains of different size. These features aid the snow-cloud discrimination. Therefore, it is a common practice to use IR channels in addition to VIS for such retrievals (Spangenberg et al., 2001). In the current task, the

limited spectral range of MERIS does not allow effective usage of this approach.

- *Image processing and spatial variability analysis* (e.g. Martins et al, 2002). In the case of white clouds over white surface, the spatial variability would mainly come from the difference in grain/particle size, surface roughness, different water phase (ice surface vs water cloud, melting surface vs ice cloud), and cloud shadows. Given the great natural variability of these parameters in both Arctic clouds and surface and the similarity of their optical properties in the given spectral range, this

approach is prone to false detections.

Combinations of the above methods together with additional thresholds and additional meteorological/reanalysis data are also available. E.g. the MODIS cloud detection scheme (Ackerman et al., 1998; Liu et al., 2004) is one of the most comprehensive among the available cloud detection schemes and is based on such combination. This algorithm uses 19 out of 36 MODIS channels along with additional inputs, e.g. topography and illumination observation geometry for each 1-km

pixel, land /water mask, ecosystem maps, and daily operational snow/ice products (taken from the NOAA and National snow and Ice Data Center). The resulting MODIS cloud mask contains 4 confidence levels (confident cloudy, uncertain, probably clear, confident clear) and is available as a separate daily averaged product. Unfortunately, due to the time lag between ENVISAT and Terra/Aqua, MODIS cloud mask product cannot be used for swathwise screening for the melt ponds fraction retrieval.

Most of the cloud screening approaches do not focus on the case of the snow surface; among those who do (Allen et al., 1990; Spangenberg et al., 2001; Trepte et al., 2001; Istomina et al., 2010; Istomina et al., 2011), even smaller fraction utilizes MERIS sensor for this task (Kokhanovsky et al., 2009, Schlundt et al., 2011, Zege et al., 2015, Istomina et al., 2015, Krijger et al, 2011).

## 2 Cloud screening for MERIS

The goal of the current work is to produce a reliable cloud screening method for MERIS data over the Arctic sea ice in summer. The currently available cloud masks for MERIS (Zege et al., 2015, Schlundt et al., 2011, etc.) are based on the





normalized indices like NDSI (Normalized Difference Snow Index) and MDSI (MERIS Differential Snow Index). In the absence of IR channels these thresholds will result in a residual cloud contamination over snow and sea ice.

However, the historic MERIS data can be collocated with the AATSR data in the center part of the MERIS swath. This AATSR data have IR channels (1.6, 3.7, 11 and 12 µm) and they can be used for training and validation of the developed MERIS cloud mask. In this work, we use the AATSR cloud screening developed for the aerosol retrieval over snow and ice (Istomina et al., 2010). This method is based on dynamic thresholds in VIS, NIR and TIR channels which discriminate snow and ice signature from all other surfaces, and from clouds.

Unlike most of the moderate resolving spectroradiometers, MERIS has the so-called oxygen A Band (MERIS channel 11 at 761.5 nm). This band can also be used to aid the cloud screening over snow and ice.

## 2.1 Oxygen A Band and the *smile* effect

As oxygen is well mixed in the Earth atmosphere, the amplitude of the absorption within MERIS channel 11 reflects optical path length of light rays received with the sensor. This band is therefore useful for cloud screening: effective path length over clouds is shorter than that over sea ice or snow on land, that is, light over higher clouds experiences less absorption when travelling through the atmosphere than light reflected from the surface. This allows separating reflecting objects such as clouds and snow/sea ice surface according to their height in the atmospheric column.

This approach has been used by Zege et al., (2015), and Istomina et al., (2015) as an additional threshold to classical whiteness and brightness criteria. For the additional threshold, the ratio of bands 10 (oxygen A reference) and 11 (oxygen A absorption) has been used R11/R10<0.27. To identify the cloud free pixels, we detect pixels where the oxygen absorbs light within the whole atmosphere column as opposed to cloudy pixels where the absorption occurs only within the small fraction of the atmosphere column, namely, above the cloud.

However, as MERIS is a push-broom sensor, its channels are susceptible to the usual for this type of sensors *smile* effect. The *smile* effect appears as characteristic along track stripes in the satellite image. It is caused by shifts of the central wavelengths of the detector's pixels. The channel 11 of MERIS λ=761.5 nm lies within the oxygen absorption band, where a slight shift in wavelength may cause drastic effect on the signal measured by the sensor. As seen in Zege et al., 2015 and Istomina et al., 2015, the channel 11 is virtually impossible to use effectively due to strong artifact presence.

Available *smile* effect corrections comprise those included into the ESA toolbox for ENVISAT processing, i.e. open source packages BEAM or SNAP (https://www.brockmann-consult.de). These corrections work well within the transparency window of the atmosphere over darker surfaces. A set of corrections produced especially for bright Arctic surface and the oxygen A band are based on the simulation of the atmospheric transmittances with a radiative transfer forward model for each given pixel with its own wavelength (Jäger, 2013). This correction gives considerable improvement on the absolute values of the measured reflectances but does not entirely remove the stripes along the swath, which hinders the usage of this correction for the cloud screening.





In this work, we suggest a *smile* correction for MERIS band 11 which allows slight inaccuracy on the absolute value of the TOA reflectances but preserves the relative difference between the sensor pixels, which allows a quantitative use of the corrected oxygen A band for cloud screening (Section 3.3.1).

## 3 Methods

The cloud screening method for MERIS data developed in this work is specifically aimed to work well over summer sea ice. It is called MECOSI (MERIS Cloud screening Over Sea Ice). Currently it is being applied as preprocessing for the retrieval of melt pond fraction and spectral albedo of summer sea ice (Melt Pond Detector, MPD). The MPD retrieval takes top-of-atmosphere reflectances of MERIS at 9 channels as input and employs a forward model of optical properties of the Arctic surface and an iterative procedure to retrieve spectral albedo and melt pond fraction of a given pixel. Several hundred field
spectra of the Arctic sea ice and melt ponds have been used to constraint the input parameters of the forward model and to ensure realistic range of modeled surfaces. More details on the MPD retrieval can be found in Zege et al. (2015). The presented cloud screening method can be used for other remote sensing applications as well, e.g. for retrievals of other surface or atmospheric parameters or as a cloud mask for coarser resolving sensors onboard same satellite platform (e.g. SCHIAMACHY on Envisat).

### 3.1 Data used

Input for MECOSI are MERIS Level 1B observations. MERIS consists of five cameras scanning the surface of the Earth in push-broom mode and offers 15 spectral bands from 412.5 nm to 900 nm. The data is collected globally with a spatial resolution of 1040×1200m at nadir. The Level 1B product provides calibrated and georeferenced top of atmosphere (TOA) radiances. These are preprocessed using the software package BEAM (www.brockmann-consult.de/cms/web/beam/).
The preprocessing includes:

1.   The region north of 65∘N is cut out from each orbit using the module Subset.

2.   The metadata in the L1B swaths is given in a grid with reduced resolution and needs to be interpolated in order to have the data available for each pixel. This is done using the BandMath module. The coordinates as well as sun zenith and the view zenith angles are now interpolated.

3.   The TOA radiances are corrected and converted to reflectances using the module Meris.CorrectRadiometry. The correction includes an equalization step to reduce detector-to-detector differences and a scheme to reduce the *smile*-effect in all but the absorption Bands 11 and 15.

A cloud mask derived from AATSR data is used as a reference mask to develop and validate the MECOSI algorithm. The AATSR instrument has been launched together with MERIS aboard ENVISAT and both sensors observe the same scene
nearly simultaneously. However, AATSR has a narrower swath of 512 km and covers only the central half of a MERIS swath. The AATSR cloud screening algorithm has been developed for an aerosol optical thickness retrieval and is presented





by Istomina et al. (2010). It exploits knowledge about the spectral shape of snow in visible, near infrared and thermal infrared bands of AATSR. The output is a binary mask for cloud free snow and ice. Validation against a number of independent datasets has proven the reliability of the algorithm in the Arctic region (Istomina et al., 2010). The training dataset used in this work was prepared as follows: all AATSR swaths from May to September 2009, 2010 and 2011 have

been subset, transformed into TOA, and co-located to the corresponding MERIS swaths using a nearest neighbour algorithm. To avoid influence of collocation errors and subpixel cloud fraction at the borders between clouds and clear sky we exclude a two-pixel border. These pixels are not used to develop or to validate the algorithm.

This AATSR dataset from May to September 2009 – 2011 was used to estimate the cloudy and clear case probabilities for given feature vector as described in the next Sections.

**3.2 Bayesian cloud screening**

A comprehensive introduction to the theory of Bayesian cloud screening is given by Hollstein et al. (2015). The described approach can be found in detail in (Marks, 2015). In the following, P(A,B) denotes the occurrence probability of A under the condition of the occurrence of B and $\mathbf{F}$ is a vector of features derived pixel-wise from satellite data. and if C denotes cloudy conditions ($\bar{\text{C}}$ – clear conditions), the probability to see a cloudy pixel under the occurrence of $\mathbf{F}$ can be written as:

$$P\left(C, \mathbf{F}\right) = \frac{P\left(\mathbf{F}, C\right) \cdot P\left(C\right)}{P\left(\mathbf{F}, C\right) \cdot P\left(C\right) + P\left(\mathbf{F}, \bar{C}\right) \cdot P\left(\bar{C}\right)} , \tag{1}$$

using this equation to calculate the cloud probability $P\left(C, \mathbf{F}\right)$ we need to estimate the probabilities $P\left(\mathbf{F}, C\right)$ and $P\left(\mathbf{F}, \bar{C}\right)$ for each possible feature vector $\mathbf{F} \in R^N$. We accomplish this by calculating *N*-dimensional frequency histograms, one for cloud and one for clear sky cases as flagged in the AATSR mask. This is done for every AATSR and MERIS swath for the time period 01.05.2009 to 30.09.2009. The background probability $P\left(C\right)$ is directly calculated from the AATSR masks using data

from the same year. Pixels outside the AATSR swath are not used in this analysis. The set of features for which the above procedure is being performed is described below.

**3.3 Features and applied corrections**

The selection of the features used to build the feature vector $\mathbf{F}$ is the most important step during the development of the algorithm and greatly affects the performance of the screening. Hollstein et al. (2015) used a random search algorithm to find a

set of features $F_i$ that performs best in global application. Here, however, the features are selected manually to find a set that performs best over snow-covered ice and darker, ponded ice. Additionally, correction algorithms were developed to equalize the systematic dependencies on the cross-track pixel position.



### 3.3.1 Oxygen-A ratio

The TOA ratio of the $O_2A$ Band 11, which is located at the oxygen absorption line at 761 nm, to Band 10 at 754 nm, which is the oxygen reference band, allows to estimate the absorption by oxygen in the atmospheric column above reflecting surface:

$$r_{ox} = \frac{R_{11}}{R_{10}},$$
(2)

As oxygen is uniformly distributed in the atmosphere, the oxygen absorption depends on the pathlength that the photons have traveled on the way from the sun to the sensor, so the ratio (2) can be used to estimate the height in the atmosphere at which the photon reflection has happened. As clouds are higher than snow and sea ice, we expect to see a decreased absorption in cloud cases. We expect this criterion to work best for optically thick water clouds. The sensitivity to optically

thin clouds is expected to be small over bright surfaces like sea ice (Preusker and Lindstrot, 2009), and clouds with a low top height would also have a weaker effect on $r_{ox}$.

The ratio $r_{ox}$ cannot be used directly in the feature vector $\mathbf{F}$ because of dependencies to the illumination-observation geometry, directional dependence of the surface optical properties (snow and sea ice BRDF), and sensor specific properties (the *smile* effect). The length of the optical path through the atmosphere depends on sun and view zenith in both cloudy and

clear cases. As these angles are provided in MERIS Level 1B swath data, the air mass factor can be calculated (e.g. Gómez-Chova et al. (2007)). However, $r_{ox}$ is strongly affected by the *smile* effect, which occurs due to a small variation in the central wavelength across the MERIS swath. The *smile* effect of MERIS has been well studied (Bourg et al., 2008) and possible ways to use this information to correct $r_{ox}$ have been shown by Gómez-Chova et al. (2007) or Jäger (2013). The approach by Jäger (2013) greatly improves the usability of $r_{ox}$, but does not fully remove detector-to-detector differences. A

reason for this might be instrument stray light, which is not fully removed in the MERIS operational processing chain (Lindstrot et al., 2010), and that was not taken into account by Jäger (2013).

In this work, we propose an empirical approach to equalize $r_{ox}$ and decrease the influence of the above-mentioned factors across the swath. We assume that over a statistically significant sample, the mean value of $r_{ox}$ for a given set of conditions (e.g., for a given detector index, geometry, etc.) can be used to correct the systematic across-track dependence for this set of

conditions. We assume that $r_{ox}$ depends on three parameters: The detector index $I_d$ which corresponds to the position of the pixel in the detector array, the reflectance at 779 nm $R_{12}$ and the sun zenith angle $\Theta_s$. $I_d$ gives a pixel's position in the sensor array and allows to compensate for the spectral *smile* effect. The dependence on $R_{12}$ is assumed to correct the influence of surface albedo and instrument stray light. It was preferred over $R_{10}$ to avoid a direct dependence on $r_{ox}$. The sun zenith angle $\Theta_s$ allows estimating the downside length of the optical path. To fully account for the acquisition geometry, the

view zenith angle $\Theta_v$ would be also required. However, we do not include $\Theta_v$ here to keep the number of dependencies as small as possible. Instead, we use $I_d$ as a proxy for $\Theta_v$ because $\Theta_v$ does not change significantly for a given detector index in the Arctic region. So, we obtain a set of data vectors:



$$M = \{(r_{ox}, \theta_s, I_d)_i\}, \; i \in I \tag{3}$$

The set $I$ denotes the indices of all pixels in one swath. Pixels with the same detector index $I_d$ are selected from the set M and corresponding subsets are built:

$$M^j = \{(r_{ox}, \theta_s, I_d) \in M \mid I_d = j\} \tag{4}$$

These subsets $M^j$ are then processed separately. The ratio is binned as follows:

$$R_\theta^j = \{r_{ox} | (r_{ox}, \theta_s, I_d) \in M^j, \; \theta \le \theta_s < \theta + \delta\} \tag{5}$$

The bin width $\delta$ is set to 1/4 degree. The sets $R_\theta^j$ are calculated for many swaths K, typically all summer data of one year. Then the mean value of $r_{ox}$ is calculated for each one of these sets:

$$\bar{r}_\theta^j = \text{mean}\{r_{ox} | r_{ox} \in \cup_k^K (R_\theta^j)_k\} \tag{6}$$

Finally, a 5$^{\text{th}}$ order polynomial is fitted to the averaged values for each separate detector index $j$ to achieve smooth and continuous correction functions $f^j$:

$$f^j = \text{fit}\{\bar{r}_\theta^j\}, \tag{7}$$

which in addition are functions of the solar zenith angle $\theta_s$. The correction is applied pixelwise by evaluating $f$ and
subtracting the resulting value from the O$_2$-A ratio. The corrected ratio is then used as a feature in the cloud screening algorithm.

It must be noted that as the further calculation of cloud probabilities for the given detector indices and values of $r_{ox}$ happens in the space of corrected $r_{ox}$ only, the absolute amplitude of $r_{ox}$ is not important for our application and is not preserved within the described approach. Instead, the relative difference between the scattering events at the surface and at the cloud
are equalized throughout the swath and thus made available for cloud screening.

The above described approach has been performed over all MERIS swathes subset to above 65°N for the time range from 01.05.2009 to 30.09.2009. This sample is considered to be a statistically significant in terms of variety of surface and cloud types and their seasonal behavior under a variety of observation-illumination geometries for all detector indices.

### 3.3.2 MERIS differential snow index

The MERIS Differential Snow Index (MDSI) is defined as normalized difference of the TOA reflectances at 865 nm and 885 nm:

$$F_{si} = \frac{R_{13} - R_{14}}{R_{13} + R_{14}}, \tag{8}$$



It exploits the drop in spectral reflectance of snow and ice at the given wavelengths to aid discrimination of snow and ice from clouds (Schlundt et al., 2011). The systematic cross-track variation is less pronounced than that for the $O_2$ -A ratio and no dependence on the observational geometry is expected, i.e. it is assumed to be the same for both spectral bands $R_{13}$ and $R_{14}$. Therefore, we use a simplified correction scheme: the mean value of $F_{si}$ is calculated for each detector index using swaths from the summer 2009. Clear sky pixels that show open water are excluded during this step. As before, to remove the systematic across-track variability, the obtained mean values are subtracted from $F_{si}$ for each detector index.

### 3.3.3 Brightness and whiteness

Many types of clouds have a higher reflectance than snow in the NIR and they usually show a white spectrum. The usefulness of these two features to detect clouds has been shown in Gómez-Chova et al. (2007) and the same definitions are used here. The brightness b is a spectral integral over the reflectance and is calculated by numerical integration of the measured TOA reflectance:

$$b = \frac{1}{\lambda_{max}-\lambda_{min}} \sum_{i \in I} \frac{r_{i+1}+r_i}{2} (\lambda_{i+1} - \lambda_i) , \qquad (9)$$

Here, $\lambda$ denotes the center wavelength of a MERIS band and $I$ is the set of used bands. The absorption bands 11 and 15 are excluded from the calculation, hence, we use I = [1, 14] \{11} to calculate the overall brightness b. The whiteness w of the spectrum is measured by the deviation of the radiances from the brightness b. With $e_i = |r_i - b|$, the equation is

$$w = \frac{1}{\lambda_{max}-\lambda_{min}} \sum_{i \in I} \frac{e_{i+1}+e_i}{2} (\lambda_{i+1} - \lambda_i) , \qquad (10)$$

Note that small values for w correspond to a flat and therefore white spectrum.

### 3.4 Evaluation

The cloud probabilities for each given set of features (Section 3.2) were compiled into binary masks in order to compare the results to the binary AATSR cloud masks. The masks are created by applying a threshold $t_p \in [0,1]$ to the cloud probability P(**F**,C) followed by one iteration of morphological closing and opening to remove isolated pixels in clear sky and cloud covered areas. Invalid pixel and clear sky open water pixel are tracked during the morphological operations to avoid an enlarged land or open water mask.

The binary MECOSI and AATSR cloud masks are the used to filter out clouds in the MPD swath data. No co-location or interpolation is necessary for this step because both algorithms, the MECOSI cloud screening and MPD, process identical MERIS swaths, and the AATSR cloud masks were gridded to the MERIS grid. The comparison of the three cloud masks, as well as illustration of separate features of the feature vector **F** as well as their corrections, is given in the next section.



## 4 Results

### 4.1 O₂A correction

An example of the influence of the O$_2$A correction described in Section 3.3.1 is presented in Figure 1. The jumps at the transition between the five detectors of MERIS, visible as vertical stripes in the uncorrected ratio (Fig. 1b), are strongly

reduced by applying the correction (Fig. 1c). The influence of low sun elevation, which causes the dark top left corner in the uncorrected ratio, is much less apparent. Also, there are no pronounced artifacts introduced by the discrete look up table (Section 3.3.1) used for the correction, as the corrected ratio is a rather smooth image. Very bright pixels, e.g. cloud edges visible in Figure 1a, are darker and more apparent after applying the correction.

Another way to investigate the effect of the correction is to study the along-track mean of the O$_2$A ratio. As expected, the

corrected ratio is a smooth function with values close to zero, if data from the whole period May to September is considered (Fig. 2 black line). This is different for the data from May only, where we find small jumps between the detectors (Fig. 2 red line). Moreover, there is a negative slope in the along-track mean, which implies that pixels at the right side of the swath tend to be darker than the ones on the left side. For the data of July, we find a reverse sign situation (Fig. 2, blue line). This seasonal dependence is expected due to the illumination-observation geometry change in the course of summer; however,

these artifacts are minimal and still allow a high-quality cloud detection using the oxygen A MERIS band.

### 4.2 Comparison to AATSR cloud mask

We first investigate whether the MECOSI algorithm can reproduce the AATSR cloud mask for the year 2009 used for the algorithm training. As AATSR data contains also TIR bands, in which the snow and ice surface is virtually a black body, the cloud detection with AATSR shows good reliability in the Arctic (Istomina et al., 2010) and can be used as a reference in

this study. Figures 3 and 4 show two examples of the MECOSI cloud probability, one for the typical situation at the beginning of the melt season in May with bright, snow covered ice (Fig. 3) and one for darker, ponded ice at the peak of the melt season in July (Fig. 4). In both cases, the cloud probability (Fig. 3b and Fig. 4b) corresponds to the AATSR mask (Fig. 3c and Fig. 4c). Most clouds visible in the TOA reflectance images (Fig. 3a and Fig. 4a) are prominent with significantly higher cloud probabilities. No distinct difference in cloud probability is visible across the swath and dependencies to the

acquisition geometry or detector specific properties appear to be well compensated. However, closer inspection reveals several cases of false negatives, like e.g. the semi-transparent clouds over landfast ice which cannot be discriminated from clear sky regions by their cloud probability (red arrow in Fig. 3a). The opposite case is shown with a blue arrow Fig. 3a, where low ice concentrations close to the coast were falsely detected as high cloud probability.

To quantify the performance of the algorithm, we study the distribution of cloud probability for clear sky and cloud covered

pixels in the AATSR mask (Fig. 5). For cloud covered pixels, we find that nearly 85% percent show a cloud probability greater than the background probability P(C)=0.86 and the distribution drops sharply towards smaller cloud probabilities (Fig. 5 top). Visual inspection shows that probabilities smaller than P(C) are almost always correlated to semi-transparent





cloud over snow covered ice or optically thin clouds. The distribution for clear sky pixels is less distinct (Fig. 5 bottom). It drops towards higher cloud probabilities, which is expected, but 6% percent show a cloud probability higher than P (C) and cannot be reliably discriminated from clouds. The majority of these 6% is the challenging case of bright, snow covered sea ice during the beginning of the melt season and fresh snow during fall freeze-up, hence such incorrectly high cloud

probability is rarely found for darker ice with melt ponds on top. Most of these false positives are connected to cloud-like values of the MDSI feature Fsi, which may potentially occur for fresh snow with fine grains. The extremely high albedo of such surface will compromise the $r_{ox}$ feature and prevent correct detection.

We compare the MECOSI binary mask to the AATSR reference mask to study the temporal behavior of the algorithm's performance and to investigate the accuracy of the binary mask. By comparing all swaths from May to September 2009, we

find that, with reference to the AATSR cloud mask, 92.51% of the MERIS pixels are classified correctly and the remaining 7.49% split up to 4.64% missed cloud and 2.85% missed clear sky pixels. The temporal behavior of the detection rates is presented in Figure 6. The algorithm works best in July, with detections rates around 0.9 for both clear sky and cloud pixels, and the performance is only slightly worse in June. However, we find a considerably worse detection rate for clear sky regions in May, August and September with values close to 0.6 and below. This indicates that more than 40% of the pixels

marked as clear sky in the AATSR mask are falsely screened out in the MECOSI binary mask. The detection rate for cloud steadily increases during June and July up to almost 1.0 at the end of the melt season. This increase is due to the state of ice surface, which gets darker over time and makes the detection of semi-transparent cloud easier.

The binary cloud mask derived from MECOSI cloud probability is compared to independent AATSR mask from two other years. By comparing over $3.8 \times 10^9$ pixels from 2010 and 2011, we find that 90.50% (90.65% for 2011) of the pixels are

correctly classified, which is about 2% less than for 2009. Thereby 5.85% (5.92%) are missed cloud and 3.64% (3.42%) are wrongly screened out clear sky pixels.

### 4.3 Extension beyond AATSR swath and comparison to MODIS cloud fraction

The accuracy of the MECOSI algorithm outside of the center half of the swath is difficult to assess because of the lack of appropriate reference data. Visual inspection of MERIS images from 2009 to 2011, which have been superimposed with the

binary cloud mask, gives the general impression that the accuracy is considerably good throughout the full swath. The several cases of semi-transparent clouds in May and early June 2010 are more frequently missed in the upper right quarter of the swath. The reason for this is somewhat small values in the corrected oxygen A ratio; a tendency towards smaller values on the right side of the swath is also observable in May 2009 (Fig. 2). The along-track mean of cloud probability for the year 2010 also gives slightly smaller values at the right side of the swath, as Figure 7 shows, and the standard deviation σ

increases. However, the differences across the swath are small (±0.017 for the mean and ±0.02 for σ) and are mainly linked to different characteristics of the five detectors of MERIS, as the jumps at the transitions and the linear behaviour for the center detectors show.





To further investigate the performance outside of the AATSR swath as well as the overall accuracy, we compare MECOSI binary cloud mask, gridded to a one-degree constant angle grid, to MODIS cloud fraction (Ackermann et al., 2008) data from May to Septmeber 2010. Thereby, we use either the full MERIS swath, center half or the outside quarters (Figure 8a,b and c, respectively). We find a good agreement with the MODIS data in all three cases. If the full MERIS swath is used (Fig. 8a), the comparison of over $6.7 \times 10^5$ grid cells gives a RMSD = 0.18 and a difference of means D = -0.02, which indicates that the MECOSI algorithm tends to retrieve slightly higher cloud fraction. The numbers for the central part of the swath (Fig. 8b) are very similar, with RMSD = 0.19 and D = -0.03, but the number of grid cells N = $5.0 \times 10^5$ is smaller because of the restricted spatial coverage. For the outside quarters, we find again almost equal parameters with RMSD = 0.19, D = -0.01 and N = $4.6 \times 10^5$, although a slight pixel displacement is seen (compare top left and bottom right corner of Fig. 8b and 8c).

**4.4 Influence on the melt pond fraction retrieval**

Finally, we study the influence of different cloud masking schemes on the retrieved MPF. Figure 9 shows an example of using the original cloud screening built into the MPD algorithm, as well as the effect of additionally applying the MECOSI and AATSR cloud masks. It is evident that both the MECOSI and the AATSR cloud mask (Fig. 9b and c) are much more restrictive than the MPD cloud masking scheme (Fig. 9a). The spatial coverage is significantly reduced and regions which are not screened out correspond well to a MODIS cloud fraction below 50% (Fig. 9d). Differences between using the MECOSI and the AATSR cloud mask are mostly due to the limited spatial coverage of AATSR (e.g. the larger pole hole).

A time series of the Arctic-wide mean MPF for all three cloud masking schemes is presented in Figure 10. The spatial coverage has been restricted to the area seen by AATSR.

For all three years 2009 to 2011, we find evident differences between the original MPD product and the two improved products with additional cloud masking. The most prominent one is the significantly higher (up to 0.08 increase) mean MPF in July when additional cloud screening is applied. In May and September, however, the additional screening results in slightly smaller mean MPF. This behavior is expected because as the MPD algorithm retrieves values of around 0.15 MPF for opaque clouds, so that immense cloud contamination in the original MPD product reduces the MPF value range of the timerseries towards this wrong MPF value.

If we focus on the differences between AATSR and MECOSI cloud mask (dark red and blue in Fig. 10), we find that both masks lead to a similar MPF timeseries. Using the MECOSI mask results in slightly higher MPF in May, which is possibly caused by some omitted clouds. The main advantage of the MECOSI cloud mask over AATSR is the larger spatial coverage of the latter (compare Fig. 9b and 9c).

**5 Discussion**

The results show that the MECOSI algorithm discriminates clouds from summer sea ice with good accuracy. With MECOSI, over 90% of the pixels are classified correctly, when compared to the AATSR reference.



Comparison to the independent MODIS daily cloud fractions shows good agreement with the developed MECOSI mask both in the center part of the MERIS swath where AATSR data are available for training, and on the outside edge of the swath (Fig. 8). There is no evidence that the quality of the algorithm performance worsens towards the edges of the swath. The variation of mean cloud probability and its standard deviation across the swath is dominated by detector-to-detector

differences and shows no change towards the edges of the swath (Fig. 7). Therefore, we conclude that the results of the comparison to AATSR cloud mask are, in general, valid for the full MERIS swath.

The quality of the MECOSI cloud mask for both clear and cloudy cases is the best in June and July, when the rapid melt onset and first pond drainage events happen on the Arctic sea ice (Fig. 6). Bright fresh snow compromise MECOSI cloud screening and lead to some false detections in May. The $O_2A$ ratio is well suited for improving the detection over fresh snow.

The proposed correction scheme equalizes the ratio reasonably well (compare Figs. 1b and 1c). However, the detector-to-detector artifacts indicate some residual influences of the spectral *smile* effect, surface albedo and instrument stray light which were not fully removed also by the proposed correction scheme.

The cloud detection rates at the end of the melting season in August/September are close to 100%. The not as good detection of clear cases might be connected to the reduced number of such scenes at the end of melting season, as humidity and

cloudiness increase, and the ice cover decreases with the minimum ice extent typically in the first weeks of September. For our specific application, i.e. retrieving surface parameters, it is important to screen out possibly all clouds as they bias the retrieval result. Wrong detections of clear cases as cloudy are less critical as this just reduces the spatial coverage of the product but does not affect the retrieved values.

Consequently, the MECOSI cloud screening improves the quality of the MPD MPF and albedo product. By reducing the

amount of cloud contamination, we find consistently higher pond fraction in the period Mid-June to Mid-August for all three years (Fig. 10). The cloud contaminated pixels are no longer used as input into the MPD retrieval and the resulting MPF dataset contains unbiased MPF and albedo values. The so improved resulting dataset can be used for further applications, such as assimilation into or validation of climate and melt pond models.

## 6 Summary

In this work, we present MECOSI, a new cloud screening routine for MERIS specifically developed for use over Arctic summer sea ice. Comparison to the independent MODIS cloud mask shows that the available summer Arctic MPF and spectral sea ice albedo product from MERIS (Zege et al., 2015; Istomina et al., 2015) are significantly cloud contaminated (compare Figs. 9a and 9d). The cloud screening method presented here has been developed to improve the quality of the MPF and albedo datasets.

The developed cloud masking routine utilizes all 15 MERIS channels and a reference AATSR cloud mask to calculate probabilities of cloudy and clear cases for a given set of features:

-    Oxygen A absorption and reference ratio (additionally corrected for *smile* effect),





- MERIS normalized difference snow index,
- brightness and whiteness criteria.

The dependencies on the illumination-observation geometry and the position of the pixel in the array of detectors, i.e. the detector index, have been accounted for as well. To calculate the cloudy and clear probabilities, a dataset of every AATSR

and MERIS swath from 01.05.2009 to 30.09.2009 have been used to ensure a representative sample of the sea ice, snow and cloud conditions.

The developed cloud mask shows a considerable improvement over the old MPD cloud mask. The quality of cloud detection of the new algorithm is close to the reference AATSR cloud mask, whereas MERIS does not have the IR channels which aid in the snow-cloud discrimination. The MECOSI cloud detection quality remains high also near the edges of the MERIS

swath where no AATSR training data were available. Comparison to the reference AATSR and independent MODIS cloud masks shows that the application of MECOSI has greatly increased the quality of the MPD products on both spatial (Fig. 9) and temporal (Fig. 10) scales.

The advantage of MECOSI over e.g. MODIS daily cloud fraction product is that it enables accurate cloud screening of swath MERIS data over snow and sea ice, which was not possible with the old version of the cloud screening used in the MPD

retrieval.

The developed cloud mask for MERIS over the summer Arctic sea ice, as well as the improved datasets of the melt pond fraction and spectral albedo for the entire MERIS operation time are available at the ftp server of the University of Bremen https://seaice.uni-bremen.de/data/meris/gridded_cldscr/.

**Acknowledgements**

The authors express gratitude to ESA and NASA for providing MERIS, AATSR and MODIS operational and higher-level products, to Brockmann Consult for providing the open source software packages BEAM and SNAP.

The authors are grateful to the anonymous reviewers and the editor for their effort and valuable comments on the manuscript. This work has been funded as a part of EU project SPICES.





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





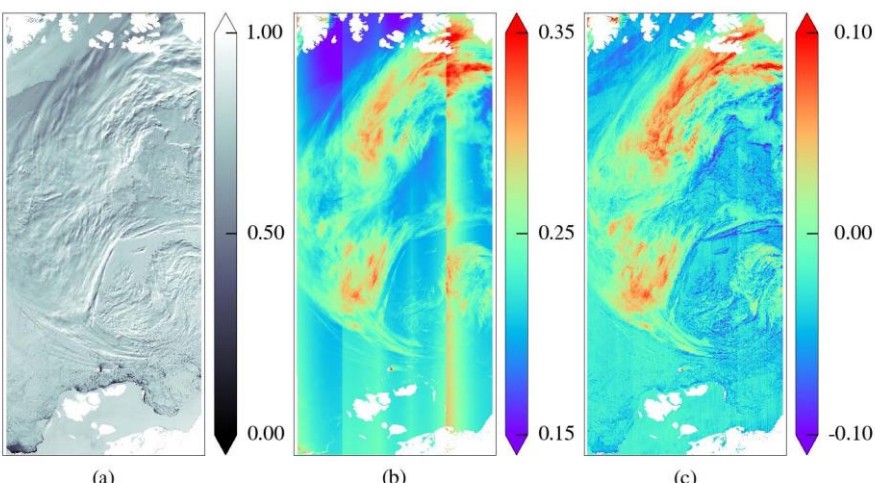

**Figure 1: Reflectance at 779 nm (a), uncorrected O₂A ratio (b) and corrected O₂A ratio used as a feature in the cloud screening (c).**
**Shown is a 2450 × 1121 pixel part of ENVISAT orbit 37475 from 1st of May 2009 with the New Siberian Islands at the bottom and parts of the Canadian Archipelago at the top. Land, open water and invalid pixels are white.**

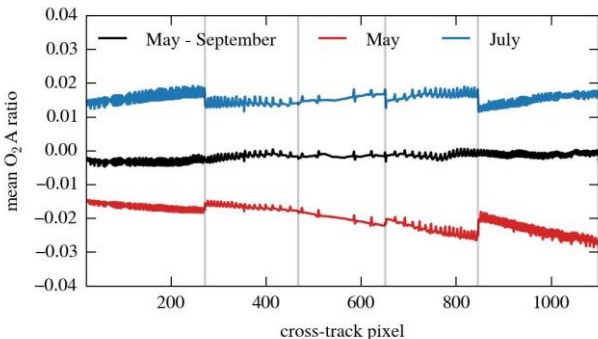

**Figure 2: Along-track mean of the corrected O₂A ratio. For each time period, the mean is calculated from 100 randomly selected swaths. The vertical lines mark the transition between the five detectors of MERIS.**





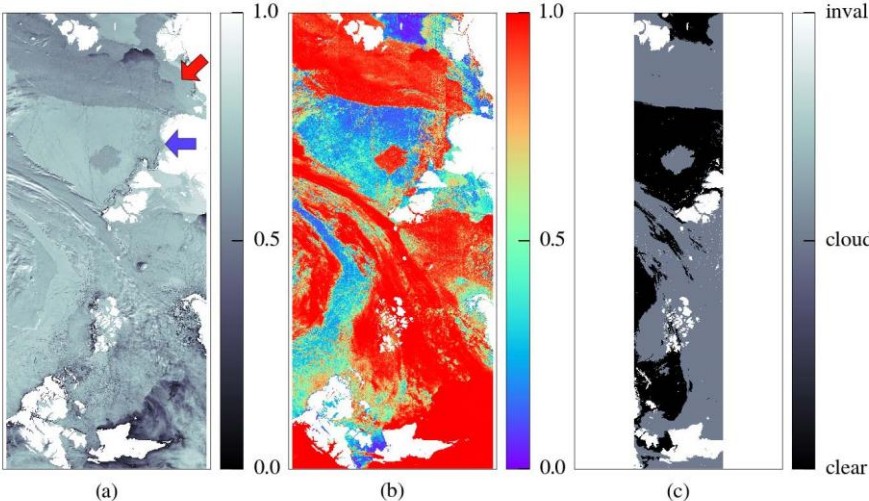

**Figure 3: Reflectance at 779 nm (a), cloud probability (b) and corresponding AATSR mask (c) for 14th of May 2009 with Svalbard at the bottom left corner. Land, open water and invalid pixels are white. The red arrow points to missed clouds and the blue one marks wrongly screened out clear sky pixels (orbit number 37666).**

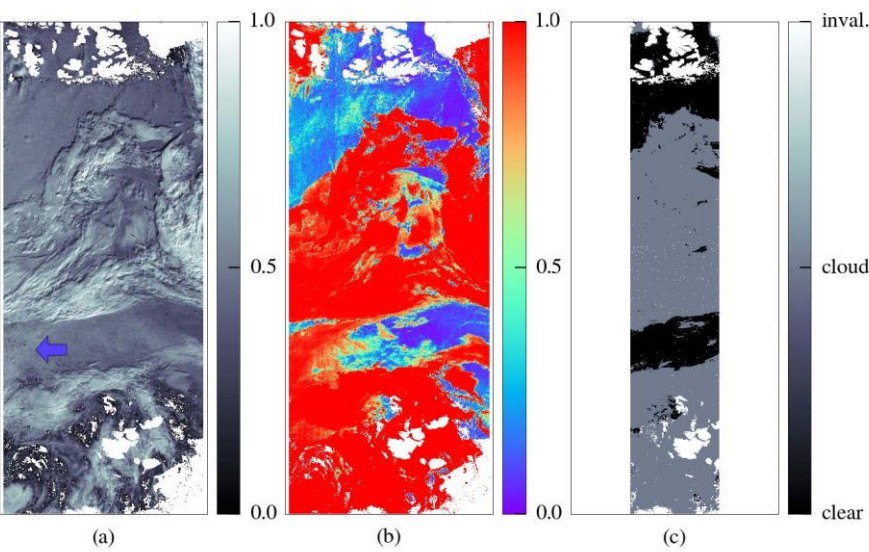

**Figure 4: Reflectance at 779 nm (a), cloud probability (b) and corresponding AATSR mask (c) for 31st of July 2009 (orbit number 38778). The blue arrow marks a region with wrongly screened out clear sky pixels, although a thin cloud cover is possible.**





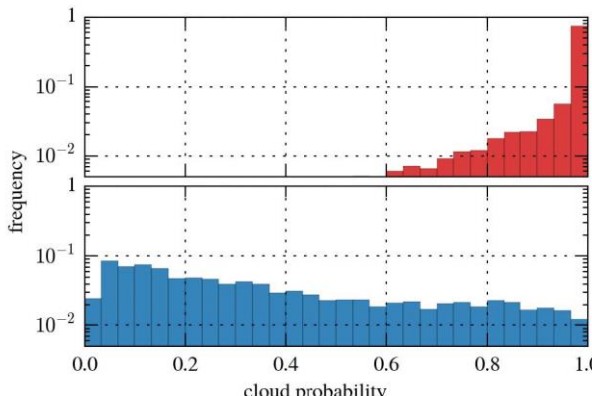

**Figure 5: Distribution of MECOSI cloud probability for AATSR cloud pixels (top) and AATSR clear sky pixels (bottom) for May to September 2009.**

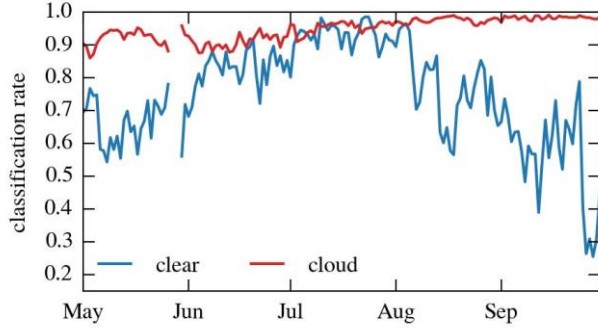

**Figure 6: Time series of daily mean classification rates for 2009. As an example, a value of 0.9 for cloud means that 90% of the cloud pixels in the AATSR mask are correctly classified as cloud covered and the remaining 10% are missed clouds.**

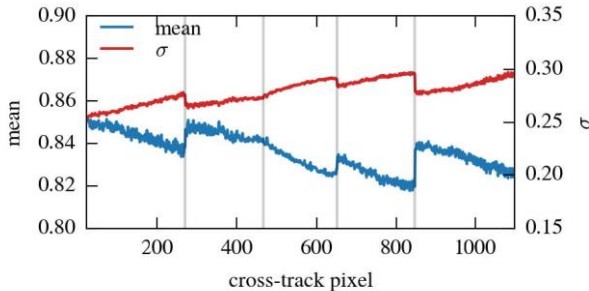

**Figure 7: Along-track mean and standard deviation of cloud probability for 2010. Vertical lines mark the transition between the**
10  **five detectors of MERIS.**





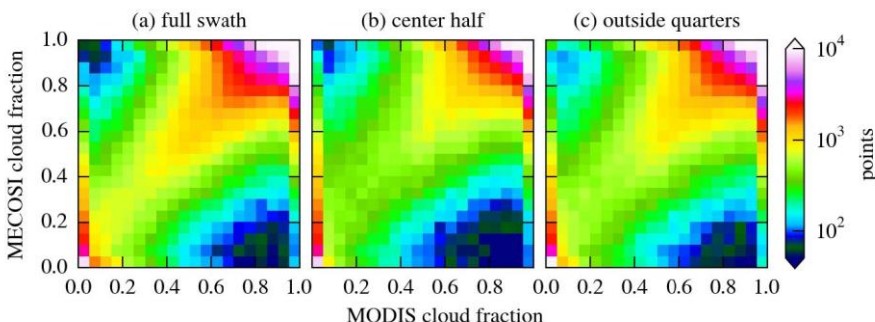

**Figure 8: Comparison of daily gridded MECOSI and MODIS cloud fraction using the full MERIS swath (a), the center half (b) or the outside quarters (c) for the gridded MECOSI fraction. Period is May to September 2010.**

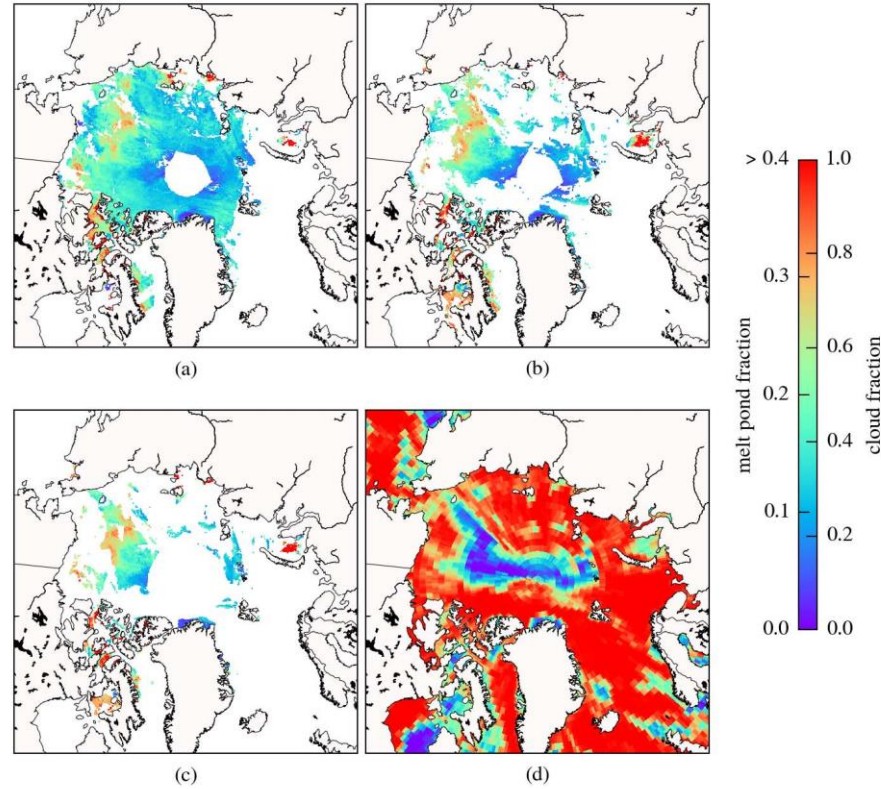

**Figure 9: Gridded melt pond fraction with MPD cloud mask (a), MECOSI cloud mask (b), AATSR cloud mask (c) and MODIS daytime mean cloud fraction (d), 20th of June 2009.**





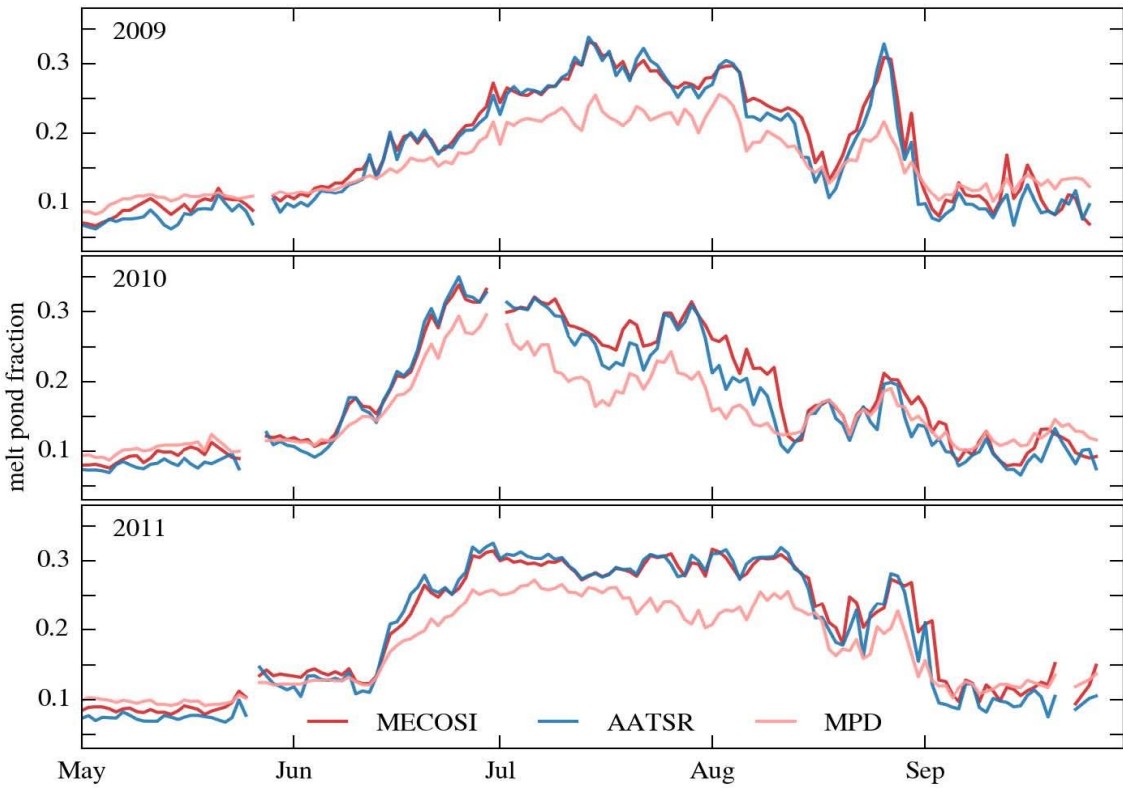

**Figure 10. Influence of different cloud mask on Arctic-wide mean melt pond fraction for 2009 – 2011. The means are calculated from gridded melt pond fraction data and coverage is restricted to the area seen by AATSR. Days with less than 100 grid cells to compare or missing AATSR data are excluded.**

