# Peer review of "Improved cloud detection over sea ice and snow during Arctic summer using MERIS data"

_Atmospheric Measurement Techniques, 2019_

## Referee Comment (RC1) · Anonymous Referee #1 · 5 Feb 2020

The manuscript describes an improved cloud detection algorithm for MERIS, developed especially for a sequential retrieval of melt pond fraction (MPF) in the summer Arctic, denoted as MECOSI. A clear improvement with respect to the previously used algorithm is demonstrated. That is, a significant progress is reported. On the other hand, the study needs to be motivated and presented more clearly.

First of all, the cloud mask from AATSR is here used as reference and is assumed to have a 100detection algorithm for MERIS seems to be an increase in the swath width for the MPF retrievals, with respect to if the cloud masking would have solely been based on AATSR. The application of retrieved MPF is not stated. If the aim is to derive climate data, I would say that close to perfect retrievals (AATSR is assumed to give perfect cloud masking) over the smaller swath is to prefer, than significantly less

accurate data over the broader swath. That is, I found the motivation to be weak, or unclear.

OLCI seems to be used as motivation in the abstract, but this sensor is not discussed at all in the text.

That AATSR should give a perfect cloud masking sounds to good to be true. The limitations of the AATSR cloud detection should be discussed. And presumably, the error of the AATSR retrievals should be considered, both when setting up the MERIS Bayesian scheme and when evaluating the performance of MECOSI.

Sections 1 and 2 needs to be restructured. At least I fail to see a clear logic in these sections. The introduction should more clearly focus on motivation and goal of the study. For example, objective/goal is now formulated in the middle of Sec. 1 and at start of Sec 2. The information around line 21 on page 1 and line 17 on page 2 is very similar, that indicates that the order is not optimal.

The review of available cloud screening approaches (Sec 1.1) is nice, but causes distraction as placed now. I would suggest to reformulate the title of Sec. 2 somewhat, and then move the review to Sec. 2.

There is a quite heavy use of acronyms, and you assume that many are understood by everybody. Note that this includes all names of satellite sensors. Is needed to use VIS and NIR? What is SGSP? Is RMSD the same RMS? MPF is defined in the abstract, but I would say that it needs to be defined in the Introduction as well.

Minor comments:

Page 4, line 18: "R11/R10<0.27" This needs further explanation.

Page 4, line 19: Writing "small fraction" is misleading as cloud systems in the Arctic typically are very shallow. In fact, are not low clouds a special problem for using oxygen A-band in this way? Probably what you mean on page 7, line 11, but this requires a more careful discussion/analysis.

Page 5: Add information about resolution of AATSR.

Page 6, line 5: What is the maximum distance of mismatch in position. That is, what is the maximum nearest neighbour interpolation?

Page 6, line 6: This sentence needs further explanation.

Start of Sec 3.3.1: Seems to be quite some repetition from Sec 2.1. Can be avoided.

Page 9, line 10: The equation below defines b as a mean, not an integrated value.

Page 9, line 14: I don't understand what "I = [1, 14] {11" means.

First paragraph of Sec 3.4: This needs further/better explanation.

СЗ

---

## Referee Comment (RC2) · Anonymous Referee #2 · 10 Feb 2020

Authors: Istomina et al. MS No.: amt-2019-413 MS Type: Research article

General comments:

The authors presented a new cloud detection method for MERIS. The method adopts the Bayesian concept with the feature vector including three parameters: O2 A-band ratio, MERIS differential snow index, and brightness and whiteness. The authors also developed a new method to correct the smile effect. It is found that the new method improves the current one significantly. The paper is relevant to the community. I recommend publication after addressing the issues listed below.

When applying the O2 A-band ratio for cloud detection, as the authors pointed out, rox is dependent on the reflectance at 779 nm, but I didn't see where this is reflected. It

[Figure]

seems the data were not binned by the 779nm reflectance. How is it taken into account in the cloud detection algorithm?

There are many acronyms not having the fully spelt version. Please check.

Specific comments:

P2, L14-16: why would "the retrievals of MPF and albedo discussed in this work misinterpret the cloud contamination as melting sea ice"? Doesn't melting sea ice have very different spectral signature with cloud?

P4, L1: Since MERIS does not have SWIR channels, how is NDSI derived?

P7, L6: Please consider changing "uniformly distributed" to "well mixed"

P9, L5: "Clear sky pixels that show open water are excluded during this step". Is there a pre-step that determines clear vs cloudy? How does this work inside the cloud detection algorithm?

––––––––––––––––––––

---

## Author Comment (AC2) · 11 Jun 2020

Please find the author's response attached as a supplement (PDF).

Please also note the supplement to this comment:
https://www.atmos-meas-tech-discuss.net/amt-2019-413/amt-2019-413-AC2-supplement.pdf

---

## Author Response (AR1)

Authors response to the comments by Anonymous Referee #1

**The manuscript describes an improved cloud detection algorithm for MERIS, developed especially for a sequential retrieval of melt pond fraction (MPF) in the summer Arctic, denoted as MECOSI. A clear improvement with respect to the previously used algorithm is demonstrated. That is, a significant progress is reported. On the other hand, the study needs to be motivated and presented more clearly.**

The authors appreciate the effort of the Anonymous Referee, the positive review and constructive comments!

**First of all, the cloud mask from AATSR is here used as reference and is assumed to have a 100detection algorithm for MERIS seems to be an increase in the swath width for the MPF retrievals, with respect to if the cloud masking would have solely been based on AATSR. The application of retrieved MPF is not stated. If the aim is to derive climate data, I would say that close to perfect retrievals (AATSR is assumed to give perfect cloud masking) over the smaller swath is to prefer, than significantly less accurate data over the broader swath. That is, I found the motivation to be weak, or unclear.**

This certainly is a valid concern. One needs to note that not only AATSR has 512km wide swath as compared to 1150km MERIS swath, but also AATSR coverage in the polar region is limited (compared AATSR and MERIS in Fig. 9 of the manuscript). That is, MERIS does provide a better global coverage and is preferable for the presented study. The motivation behind is twofold:

1.      To the knowledge of the authors, at the time of writing no climate model includes melt ponds on top of sea ice. Although field measurements of melt ponds have been performed and published since a long time, i.e. an assimilation into a climate model within a limited spatial range as the referee suggests would have been long possible, melt ponds nevertheless present a challenge for climate modeling due to unknown global spatial distribution. Although air temperature at the surface is available also over sea ice covered Arctic ocean, melt pond fraction is not linearly linked to the air temperature but also depends on the ice topography and its internal macrophysical properties as density, porosity etc. Satellite datasets of possibly global coverage help understand not only local events but spatial dynamics in general, which may eventually lead to successful inclusion of melt ponds into climate models.

2.      Although most of the field campaigns and in situ measurements of the sea ice covered Arctic ocean are available during Arctic summer, the links and feedbacks between the rapidly evolving sea ice surface, the atmosphere and the underice ecosystem are not yet fully understood. The appearance of melt ponds on sea ice during melt onset causes a drastic change of its albedo and transmittance which affects the surface energy balance and facilitates lateral, top, bottom and internal sea ice melt, i.e. affects the sea ice volume. Only recently the suggestion that melt ponds during melt onset might be connected to the sea ice area during the sea ice minimum has been published (Schröder et al., 2014). In order to understand these processes, a long-term global coverage record of sea ice parameters, among others also melt pond fraction, needs to be available to the community. That is, the presented cloud screening routine and the resulting melt pond fraction dataset can be used in independent studies of sea ice processes and not only in climate models.

The corresponding explanation and motivation are added into the Introduction to the text, see P2 L12-30 of the new version of the text.

**OLCI seems to be used as motivation in the abstract, but this sensor is not discussed at all in the text.**

As both sensors MERIS and OLCI are similar with OLCI being a successor of MERIS, OLCI is mentioned as means to provide a long-term melt pond fraction data record as continuation to that of MERIS. However, the presented cloud screening method has been developed specifically for MERIS sensor and the authors like to highlight that the general problem of cloud screening over snow for ENVISAT sensors, e.g. SCHIAMACHY (see e.g. Schlundt et al., 2011), has now been updated and advanced.

The corresponding explanation is added in the new version of the manuscript P3, L1-8.

**That AATSR should give a perfect cloud masking sounds to good to be true. The limitations of the AATSR cloud detection should be discussed. And presumably, the error of the AATSR retrievals should be considered, both when setting up the MERIS Bayesian scheme and when evaluating the performance of MECOSI.**

Indeed, no cloud screening routine is 100% reliable. The AATSR cloud mask, its limitations and validation are presented by Istomina et al. (2010). They highlight the challenge of cloud screening validation, with in situ point measurements (e.g. lidars) being precise but giving very limited spatial coverage, and with comparisons to other cloud masks being compromised by the time difference between the satellite overflights. The comparison of the AATSR cloud mask to the lidar data has proven its robustness (95% correct cloudy/clear detections with remaining 5% of cases connected to thin clouds on a sample of ~100 scenes).

The authors agree that this has not been addressed enough in the manuscript and add the corresponding explanation into the text, P 8, L1-4.

**Sections 1 and 2 needs to be restructured. At least I fail to see a clear logic in these sections. The introduction should more clearly focus on motivation and goal of the study. For example, objective/goal is now formulated in the middle of Sec. 1 and at start of Sec 2. The information around line 21 on page 1 and line 17 on page 2 is very similar, that indicates that the order is not optimal.**

**The review of available cloud screening approaches (Sec 1.1) is nice, but causes distraction as placed now. I would suggest to reformulate the title of Sec. 2 somewhat, and then move the review to Sec. 2.**

The authors are grateful for this comment and agree that the manuscript can be better structured. In the new version of the manuscript, we take special care to avoid repetitions and keep the text concise and clearly structured, the Sections 1 and 2 have been reformulated as suggested, see P3 L 23 onwards.

**There is a quite heavy use of acronyms, and you assume that many are understood by everybody. Note that this includes all names of satellite sensors. Is needed to use VIS and NIR? What is SGSP? Is RMSD the same RMS? MPF is defined in the abstract, but I would say that it needs to be defined in the Introduction as well.**

This problem has also been highlighted by the second referee and the authors agree that the usage of the acronyms has to be reconsidered. In the new version of the manuscript, we define MPF also in the abstract, and take care to spell out all the remaining acronyms. VIS and NIR are removed.

The corresponding changes have been added throughout the text.

**Minor comments:**

**Page 4, line 18: "R11/R10<0.27" This needs further explanation.**

This is a manually derived threshold which stems from the visual analysis of several dozen of MERIS scenes and was described and used in Zege et al 2015 (Eq 17 therein). The corresponding explanation and reference are added into the new version of the manuscript P6, L8-9.

**Page 4, line 19: Writing "small fraction" is misleading as cloud systems in the Arctic typically are very shallow. In fact, are not low clouds a special problem for using oxygen A-band in this way? Probably what you mean on page 7, line 11, but this requires a more careful discussion/analysis.**

Of course, what is meant here is "short path length" and not "small fraction". This has been corrected in the new version of the text P6, L1.

Also, the following text has been added on P5, L25-28: "and clouds with a low top height would generally also have a weaker effect onto the oxygen ratio. Fortunately, as in our case the Arctic sea ice surface lies uniformly at sea level and displays no relief, there is no confusion possible between clouds and surface in the terms of optical path length and the only uncertainty might come from the sensor specific features, i.e. the smile effect."

**Page 5: Add information about resolution of AATSR.**

The text "The spatial resolution of AATSR is 1km at nadir." is added on P7, L29.

**Page 6, line 5: What is the maximum distance of mismatch in position. That is, what is the maximum nearest neighbour interpolation?**

The mentioned here regridding has been done with the python package pyresample. The radius of influence for the nearest neighbour interpolation is 1.5km. This value has been added to the text, P8, L9.

**Page 6, line 6: This sentence needs further explanation.**

The following text has been added as explanation: "As the AATSR and MERIS data have different spatial resolution, the two datasets have been gridded to a single grid (the coarser grid of MERIS). This might have affected the pixels at the borders of clouds in a way that earlier fully covered pixels now become partly covered which the binary AATSR cloud mask cannot fully reflect. Therefore we exclude the 2 pixel border from the study." P8, L9-12.

**Start of Sec 3.3.1: Seems to be quite some repetition from Sec 2.1. Can be avoided.**

Indeed, the authors agree that the beginning of Sec. 3.3.1 has already been mentioned in Sec 2.1The sections 2.1 and 3.3.1 have been now restructured correspondingly. Text on P9, L12-28 has been partially moved to Sec 2.1.

**Page 9, line 10: The equation below defines b as a mean, not an integrated value.**

For the sake of clarity, the sentence at Page 9 line 10 has been changed correspondingly:

"The brightness b is a spectral integral over the reflectance. As the spectral resolution of the sensor is quite coarse with only 13 used channels, the brightness can be represented by the following equation:" P11, L26-27.

**Page 9, line 14: I don't understand what "I = [1, 14] {11" means.**

The authors meant "in ascending order from 1 to 14 except for 11". As the same is basically said in words in the corresponding sentence, this equation is obsolete and for the sake of clarity is removed in the new version of the manuscript. P12, L2.

**First paragraph of Sec 3.4: This needs further/better explanation.**

The first paragraph of Sec. 3.4 has been rewritten, with the following text as a substitute:

"The cloud probabilities for each given set of features (Section 3.2) were compiled into binary masks in order to compare the results to the binary AATSR masks. The masks are created by normalizing the cloud probability P($\mathbf{F}$,C) to the range [0,1] and splitting the dataset at a probability threshold 0.45 to introduce binary values. An operation of morphological closing and opening was then applied to the cloud and snow/ice pixels in order to remove single pixels." P12, L8-12.

Authors response to the comments by Anonymous Referee #2

**General comments:**

**The authors presented a new cloud detection method for MERIS. The method adopts the Bayesian concept with the feature vector including three parameters: O2 A-band ratio, MERIS differential snow index, and brightness and whiteness. The authors also developed a new method to correct the smile effect. It is found that the new method improves the current one significantly. The paper is relevant to the community. I recommend publication after addressing the issues listed below.**

The authors are grateful for the positive review and appreciate the effort of the reviewer!

**When applying the O2 A-band ratio for cloud detection, as the authors pointed out, rox is dependent on the reflectance at 779 nm, but I didn't see where this is reflected. It seems the data were not binned by the 779nm reflectance. How is it taken into account in the cloud detection algorithm?**

The reviewer has probably meant the text on Page 7 Line 26-29 of the old version. In this part, we present the correction of the systematic offset due to the smile effect and not yet the cloud screening itself. The text on Page 7 Line 26-29 describes the dependencies of $r_{ox}$ which were considered to try and remove the systematic offset so that $r_{ox}$ can be further used to derive cloud probabilities. The authors thank the reviewer for noticing this writing mishap. This part of the text stems from a draft version of the manuscript and needs to be updated. Indeed, the dependence on 779nm reflectance has not been considered in the current version of the manuscript, as can also be seen in the following equations. The problem with taking the surface reflectance into account by using the 779nm channel lies in the fact that the statistical majority of cases where the correction has to be performed is located in a relatively narrow range of surface reflectances (corresponding to wet ice/bare ice with ~20% melt ponds, a widespread situation during Arctic summer), which would correspond to only one or two bins when binning over 779nm. The sample size for the other bins (darker or brighter surface types) is orders of magnitude smaller, unevenly distributed and is not sufficient to develop a statistical correction. This can be seen in Fig. 2 where a discrepancy

of less than 2% of $r_{ox}$ value is shown when comparing corrections for the entire summer (black curve) with averaged May or July (red and blue curves). One can say, the dependence on the surface albedo is not so pronounced for the $r_{ox}$ ratio, which of course is only valid for the ratio and would not be the case for a single oxygen absorption band R11. However, as our purpose was the relative correction of the smile effect for effective usage of cloud screening thresholds and not (as e.g. in Jäger, 2013) an absolute calibration of the R11 reflectance distorted by the smile effect, the achieved accuracy of the correction of a few percent (as shown in Fig. 2) justifies the selected approach.

In turn, the sum of the solar and viewing zenith angles turned out to give a better reflection of the daily cycle in comparison to the detector index and solar angle alone, so that the viewing angle has been included into the correction scheme.

In the new version of the manuscript, the text at Page 7 lines 25-32 (old version) or P10, L1-9 (new version).

was therefore updated as follows:

"We assume that $r_{ox}$ depends on three parameters: the detector index $I_d$ which corresponds to the position of the pixel in the detector array, the sun zenith angle $\theta_s$ and the viewing zenith angle $\theta_v$. $I_d$ gives a pixel's position in the sensor array and allows to compensate for the spectral *smile* effect. The sun zenith angle $\theta_s$ and the viewing angle $\theta_v$ allows estimating the optical path in the atmosphere which is in direct dependence with the oxygen absorption. The seasonal nature of $r_{ox}$ dependence on surface reflectance e.g. at channel 779nm presents a challenge of statistically non-uniform bins of very different sample size and was not included into the correction scheme. The residual rox dependence on the surface reflectance is less than 2% (Fig. 2) and does not prevent the application of the cloud screening routine."

The indices in Eq 3-54 were corrected to $\theta_{sum} = \theta_s + \theta_v$

**There are many acronyms not having the fully spelt version. Please check.**

This is also the point highlighted by the other reviewer and the new version of the manuscript has fewer and clearly defined acronyms.

The corresponding changes have been added throughout the text.

**Specific comments:**

**P2, L14-16: why would "the retrievals of MPF and albedo discussed in this work misinterpret the cloud contamination as melting sea ice"? Doesn't melting sea ice have very different spectral signature with cloud?**

The melting sea ice displays a variety of spectral behaviors in the entire range from white ice to dark melt ponds (e.g. see Istomina et al., 2012, PANGAEA dataset of sea ice spectral albedo during Arctic summer). The specifics of the MPF and albedo retrieval is such that not only this large range of surfaces but also their subpixel mixtures in various fractions have to be represented. This requires a versatile forward model and retrieval which can account for sea ice variability at a global spatial scale (see Zege et al 2015). In the given spectral range of the MERIS (412.5 - 900nm) clouds do not differ from the variety of surfaces available during Arctic summer to the point of clear distinction. So that e.g. warm water clouds look similar to white ice throughout most of the available spectral range (same for cirrus and fresh fine snow). This results in the fact that the retrieval does confuse their reflectances and relies on additional cloud screening.

The corresponding explanation has been added in the text, P3, L11-18.

**P4, L1: Since MERIS does not have SWIR channels, how is NDSI derived?**

What is meant here is that the NDSI-like threshold is used, in this case the MDSI - MERIS Differential Snow Index. It is derived using two channels (865nm and 885nm) and utilizes the specific grain size feature of snow which is absent in other surfaces. The MDSI of this kind has been used e.g. by Schlundt et al (2011) and is also used in the presented work (see Eq. 8)

The corresponding sentence has been corrected on P5, L7-8.: "The currently available cloud masks for MERIS (Zege et al., 2015, Schlundt et al., 2011, etc.) are based on NDSI-like (Normalized Difference Snow Index) indices, e.g. MDSI (MERIS Differential Snow Index)."

**P7, L6: Please consider changing "uniformly distributed" to "well mixed"**

Thank you for this remark, the text has been changed accordingly. P5, L19

**P9, L5: "Clear sky pixels that show open water are excluded during this step". Is there a pre-step that determines clear vs cloudy? How does this work inside the cloud detection algorithm?**

Indeed, there is a pre-step that removes all open water pixels from the correction dataset. The latter is done as described by Schundt et al (2011) using thresholds on reflectances at channels 12 and 13, with the threshold values 0.09 and 0.08, respectively. The result is that the correction values for each detector index are then produced excluding the dark pixels. There is no distinction of clear and cloudy at this point as it is only the step to remove the systematic across-track variability so that the MDSI feature is not affected by it, and we can apply this feature more effectively.

The sentence: "Open water pixels have been removed using two thresholds on channels 12 and 13 as described by Schlundt et al. (2011)" has been added into the text as clarification P11, L20-21.

[revised manuscript text omitted]

For MERIS data with a spectral range from 412.5nm to 900nm, cloud detection over snow and sea ice a challenging task. However, the advantage of MERIS - its 15 spectral bands within this relatively small spectral range - makes it especially suitable for the melt pond fraction (MPF) retrieval over the Arctic sea ice, which needs a quality cloud screening routine. Although most of the field campaigns and *in situ* measurements of the sea ice covered Arctic ocean are performed during
15 Arctic summer (e.g. an overview in Istomina et al., 2015), the links and feedbacks between the rapidly evolving sea ice surface, the atmosphere and the underice ecosystem are multifold (Curry et al., 1996) and not yet fully understood. The appearance of melt ponds on sea ice during melt onset causes a drastic change of its albedo and transmittance (Nicolaus et al., 2012) which affects the surface energy balance and facilitates lateral, top, bottom and internal sea ice melt, i.e. affects the sea ice volume. Only recently the suggestion that melt ponds during melt onset might be connected to the sea ice area during
20 the sea ice minimum has been published (Schröder et al., 2014). In order to understand these processes, a long-term global coverage record of sea ice parameters, among others also MPF, needs to be available to the community. That is, the presented cloud screening routine and the resulting MPF dataset can be used in studies of sea ice processes and feedbacks. To the knowledge of the authors, at the time of writing no climate model includes melt ponds on top of sea ice. One of the reasons is that melt ponds, although observed *in situ* during many campaigns, still present a challenge for climate modeling
25 due to unknown global spatial distribution. Although reanalysis air temperature at the surface is available also over sea ice covered Arctic ocean (e.g. Kalnay et al., 1996), MPF is not linearly linked to the air temperature but also depends on the ice topography and its internal macrophysical properties as density, porosity etc. Satellite MPF datasets of possibly global coverage are the only way to understand not only local events but also global spatial dynamics, which may eventually lead to successful inclusion of melt ponds into climate models.
30 Besides cloud screening for the MPF retrieval using MERIS data,  a robust cloud detection from MERIS in the Arctic region may be important for 1) synergy with the other sensors onboard Envisat and 2) might

be applicable to sensors similar to MERIS, e.g. OLCI. , e.g. as an accurate cloud fraction for the hyperspectral sensor of coarser spatial resolution SCIAMACHY.

The cloud screening for OLCI, which is a successor of MERIS without thermal infrared bands, presents challenges similar to those of MERIS. OLCI data are important as continuation of MERIS in order to provide long-term data records of e.g. MPF. Nevertheles, the cloud screening presented here has been developed specifically for MERIS and thus addresses the issue of cloud screening over snow for ENVISAT sensors, e.g. SCIAMACHY (Scanning Imaging Absorption Spectrometer for atmospheric Chartography) (see e.g. Schlundt et al., 2011). Of course, the approach presented here can be applied to OLCI data as well.

Depending on the retrieved parameter and sensor, the effect of a compromised cloud screening may be moderate (retrievals of albedo and snow grain size within, SGSP (Snow Grain Size and Pollution Amount Retrieval), Wiebe et al., 2013) to drastic (aerosol retrieval, Istomina et al., 2011; MPF melt pond fraction retrieval, Zege et al. 2015). As the melting sea ice displays a variety of spectral behaviors in the entire range from white ice to dark melt ponds (e.g. Istomina et al., 2013), a versatile forward model and retrieval which can account for such a variability at a global spatial scale are needed. Such a retrieval (Melt Pond Detector, MPD) has been developed by Zege et al., (2015). The MPD is a pixelwise retrieval and only utilizes the spectral information without additional morphological or statistical criteria. As clouds do not spectrally differ from most of the surfaces available during Arctic summer, so that e.g. warm water clouds may appear similar to white ice throughout most of the available spectral range (same for cirrus and fresh fine snow), tThe retrievals of MPF and albedoMPD can thereforediscussed in this work misinterpret the cloud contamination as sea ice melt. as melting sea ice surface which cannot be distinguished from the true melting surface and overlays the true values in the daily and weekly means. The resulting MPF and albedo datasets are thus strongly affected by the residual cloud contamination. The objective of this work is to resolve this issue by means of a better reliable cloud discrimination over snow for MERIS and to provide the datasets of MPF, albedo and cloud mask datasets of a better quality than currently available.

**2 Cloud screening for MERISSensor specific cloud screening in remote sensing**

**1.1 Available cloud screening approaches**

[revised manuscript text omitted]

A cloud mask derived from AATSR data is used as a reference mask to develop and validate the MECOSI algorithm. The AATSR instrument has been launched together with MERIS aboard ENVISAT Envisat and both sensors observe the same scene nearly simultaneously. The spatial resolution of AATSR is 1km at nadir which is similar to the spatial resolution of

30 MERIS. However, as AATSR has a narrower swath of 512 km, it and covers only the central half of a MERIS swath. The AATSR cloud screening algorithm has been developed for an aerosol optical thickness retrieval and is presented by Istomina et al. (2010). It exploits knowledge about the spectral shape of snow in visible, near infrared and thermal infrared bands of

AATSR. ~~As intercomparisons of cloud screening routines are challenging due to the time difference between the overflights of different satellite sensors, the validation has been performed against *in situ* lidar data. The comparison of the AATSR cloud mask to the Micro Pulse Lidar data has proven the robustness of the method (95% correct cloudy/clear detections with remaining 5% of cases connected to thin clouds on a sample of ~100 scenes).~~The output is a binary mask for cloud free snow

5  and ice

The training dataset used in this work was prepared as follows: all AATSR swaths from May to September 2009, 2010 and 2011 have been subset, transformed into TOA, and co-located to the corresponding MERIS swaths using a nearest neighbour algorithm (radius of influence 1.5 km). As the AATSR and MERIS data have different spatial resolution, the two datasets

10  have been gridded to a single grid (the coarser grid of MERIS). This might have affected the pixels at the borders of clouds in a way that earlier fully covered pixels now become partly covered which the binary AATSR cloud mask cannot fully reflect. Therefore we exclude the two pixel border from the study.

[revised manuscript text omitted]

conditions. We assume that $r_{ox}$ depends on three parameters: the detector index $I_d$ which corresponds to the position of the pixel in the detector array, the sun zenith angle $\theta_s$ and the viewing zenith angle $\theta_v$. $I_d$ gives a pixel's position in the sensor array and allows to compensate for the spectral *smile* effect. The sun zenith angle $\theta_s$ and the viewing angle $\theta_v$ allows estimating the optical path in the atmosphere which is in direct dependence with the oxygen absorption. The seasonal nature of $r_{ox}$ dependence on surface reflectance e.g. at channel 779nm presents a challenge of statistically non-uniform bins of vastly different sample size and was not included into the correction scheme. The residual $r_{ox}$ dependence on the surface reflectance is less than 2% (Fig. 2) and does not prevent the application of the cloud screening routine. Assuming $\theta_{sum} = \theta_s + \theta_v$, we obtain a set of data vectors:   *I*

   *smile*

$$M = \{(r_{ox}, \theta_{sum}, I_d)_i\}, \; i \in I$$

(3)

The set $I$ denotes the indices of all pixels in one swath. Pixels with the same detector index $I_d$ are selected from the set M and corresponding subsets are built:

$$M^j = \{(r_{ox}, \theta_{sum}, I_d) \in M \mid I_d = j\}$$

(4)

These subsets $M^j$ are then processed separately. The ratio is binned as follows:

$$R_\theta^j = \{r_{ox} \mid (r_{ox}, \theta_{sum}, I_d) \in M^j, \; \theta \leq \theta_{sum} < \theta + \delta\}$$

(5)

The bin width $\delta$ is set to 1/4 degree. The sets $R_\theta^j$ are calculated for many swaths K, typically all summer data of one year. Then the mean value of $r_{ox}$ is calculated for each one of these sets:

$$\bar{r}_\theta^j = \text{mean}\{r_{ox} \mid r_{ox} \in \cup_k^K (R_\theta^j)_k\}$$

(6)

Finally, a 5$^{th}$ order polynomial is fitted to the averaged values for each separate detector index $j$ to achieve smooth and continuous correction functions $f^j$:

$$f^j = \text{fit } \{\bar{r}_\theta^j\},\tag{7}$$

which in addition are functions of  $\theta_{sum}$. The correction is applied pixelwise by evaluating $f$ and subtracting the resulting value from the O$_2$-A ratio. The corrected ratio is then used as a feature in the cloud screening algorithm.

5 It must be noted that as the further calculation of cloud probabilities for the given detector indices and values of $r_{ox}$ happens in the space of corrected $r_{ox}$ only, the absolute amplitude of $r_{ox}$ is not important for our application and is not preserved within the described approach. Instead, the relative difference between the scattering events at the surface and at the cloud are equalized throughout the swath and thus made available for cloud screening.

The above described approach has been performed over all MERIS swathes subset to above 65°N for the time range from 10 01.05.2009 to 30.09.2009. This sample is considered to be a statistically significant in terms of variety of surface and cloud types and their seasonal behavior under a variety of observation-illumination geometries for all detector indices.

**3.3.2 MERIS differential snow index**

The MERIS Differential Snow Index (MDSI) is defined as normalized difference of the TOA reflectances at 865 nm and 885 nm:

15 $$F_{si} = \frac{R_{13} - R_{14}}{R_{13} + R_{14}},\tag{8}$$

It exploits the drop in spectral reflectance of snow and ice at the given wavelengths to aid discrimination of snow and ice from clouds (Schlundt et al., 2011). The systematic cross-track variation is less pronounced than that for the O$_2$ -A ratio and no dependence on the observational geometry is expected, i.e. it is assumed to be the same for both spectral bands $R_{13}$ and $R_{14}$. Therefore, we use a simplified correction scheme: the mean value of $F_{si}$ is calculated for each detector index using 20 swaths from the summer 2009.  Open water pixels have been removed using two thresholds on channels 12 and 13 as described by Schlundt et al. (2011). As before, to remove the systematic across-track variability, the obtained mean values are subtracted from $F_{si}$ for each detector index.

**3.3.3 Brightness and whiteness**

Many types of clouds have a higher reflectance than snow in the  near infrared and they usually show a white spectrum. 25 The usefulness of these two features to detect clouds has been shown in Gómez-Chova et al. (2007) and the same definitions are used here. The brightness b is a spectral integral over the reflectance. As the spectral resolution of the sensor is quite coarse with only 13 used channels, the brightness can be represented by the following equation::

$$b = \frac{1}{\lambda_{max} - \lambda_{min}} \sum_{i \in I} \frac{r_{i+1} + r_i}{2} (\lambda_{i+1} - \lambda_i),\tag{9}$$

Here, λ denotes the center wavelength of a MERIS band and *I* is the set of used bands. The absorption bands 11 and 15 are excluded from the calculation, hence, we use  1 - 10 and 12 - 14 to calculate the overall brightness b. The whiteness w of the spectrum is measured by the deviation of the radiances from the brightness b. With $e_i = |r_i - b|$, the equation is

$$w = \frac{1}{\lambda_{max} - \lambda_{min}} \sum_{i \in I} \frac{e_{i+1} + e_i}{2} (\lambda_{i+1} - \lambda_i) ,$$

(10)

Note that small values for w correspond to a flat and therefore white spectrum.

**3.4 Evaluation**

The cloud probabilities for each given set of features (Section 3.2) were compiled into binary masks in order to compare the results to the binary AATSR cloud masks. The masks are created by normalizing the cloud probability P(F,C) to the range

10 [0,1] and splitting the dataset at a probability threshold 0.45 to introduce binary values. An operation of morphological closing and opening was then applied to the cloud and snow/ice pixels in order to remove single pixels.

[revised manuscript text omitted]